# An Adaptive Sampling Period Approach for Management of IoT Energy Consumption: Case Study Approach

**DOI:** 10.3390/s22041472

**Published:** 2022-02-14

**Authors:** Carlos Rodriguez-Pabon, Guillermo Riva, Carlos Zerbini, Juan Ruiz-Rosero, Gustavo Ramirez-Gonzalez, Juan Carlos Corrales

**Affiliations:** 1Departamento de Telemática, Universidad del Cauca, Calle 5, No. 4-70, Popayan 190002, Colombia; gramirez@unicauca.edu.co (G.R.-G.); jcorral@unicauca.edu.co (J.C.C.); 2Grupo de Investigacion y Transferencia en Electronica Avanzada (GInTEA), Universidad Tecnologica Nacional, Griva, Cordoba C1041AAJ, Argentina; griva@frc.utn.edu.ar (G.R.); czerbini@frc.utn.edu.ar (C.Z.); 3Technology Innovation Institute (TII), Abu Dhabi P.O. Box 9639, United Arab Emirates; juanpablo.ruiz@tii.ae

**Keywords:** Internet of Things, energy efficiency, agricultural value chain, Colombian coffee

## Abstract

The Internet of Things (IoT) opens opportunities to monitor, optimize, and automate processes into the Agricultural Value Chains (AVC). However, challenges remain in terms of energy consumption. In this paper, we assessed the impact of environmental variables in AVC based on the most influential variables. We developed an adaptive sampling period method to save IoT device energy and to maintain the ideal sensing quality based on these variables, particularly for temperature and humidity monitoring. The evaluation on real scenarios (Coffee Crop) shows that the suggested adaptive algorithm can reduce the current consumption up to 11% compared with a traditional fixed-rate approach, while preserving the accuracy of the data.

## 1. Introduction

The Internet of Things (IoT) is a technological paradigm consisting of a global network between devices, which can communicate with other objects [1]. IoT opens opportunities beyond optimization and automation of processes when using the collected data. These data are an input for smart systems, which, at the same time, can provide predictions and recommendations [2,3], particularly in monitoring the agricultural value chain (AVC). Therefore, several IoT applications to the agricultural domain have demonstrated that the IoT can support planning or decision-making by owners, managers, and heads of agriculture companies [4]. As a consequence of this support, IoT brings to agriculture some advantages:Minimizing the effort required along with different phases of the agricultural value chain;Enhancing quality of the final product;Providing more information to end-user like traceability, product quality, and others;Determining the status of the product and its market demand.

However, despite the IoT benefits, there are persisting challenges in energy consumption, communication, storage security, device standardization, and data storage capacity [5,6,7,8]. These challenges create difficulties in implementing the monitoring process in AVCs, particularly in the production and post-harvest phases since monitoring variables are under outdoor conditions. Additionally, the storage and transport processes that take place over several days, weeks, or months require efficient monitoring since the environmental conditions necessary to ensure product’s quality [9].

These phases have high energy consumption levels on IoT devices, limiting the possibility of recharging them or being connected to the grid or photo-voltaic systems under outdoor conditions. Although the latter is indeed the most used solution [10], these are not applicable or represent a high cost, especially for small stakeholder groups. Therefore, managing energy consumption and device efficiency are challenges for supporting IoT-based monitoring at some stages of the AVC.

Low consumption solutions for IoT-based monitoring devices have been classified in different schemes to achieve energy conservation: Sleep/Awake [10,11], querying [12,13], hierarchical processing [14], and routing schemes [15,16,17,18]. Some of these schemes employ a few adaptive steps by filtering data, physical reconfiguration, and spatio-temporal correlation [19]. Those schemes are performed over sensor nodes and their environments under a scenario with high device density and servers. Others schemes combine clustering, probabilistic models [20,21], and frequency analysis on sensors [22,23] by implementing an algorithm in the device that allows autonomous management without operating a central node or server. Additionally, [24] uses adaptive schemes to recover data into the context of Bayesian theory; [25] proposes an adaptive sample scheduling mechanism based on signal sparsity and change intensity; [26] proposes three different adaptive sampling rate techniques for industrial wireless sensor networks (IWSNs), which can dynamically estimate the sampling frequency of the collected data; all of them focus on the difference in variation of data between samples.

However, these studies have not presented an energy conservation method in monitoring devices considering the variance of the monitored variables. This paper develops an adaptive sampling period method to save the IoT device energy and maintain the ideal sensing quality based on the variance of variables, particularly for the monitoring of temperature and humidity, which are essential for different stages in coffee AVC. The method was validated in a Case Study in a Colombian Coffee Smart Farm.

The main contributions of this paper include:A base method generally applicable to any environmental variable focused on three steps (Capture, analysis, obtain best values);An analysis that uses temperature and humidity data, with machine learning tools, finds the proposed equation’s best values;An evaluation in a real environment of the results generated from data analysis with proposed method.

The remainder of this paper is organized as follows. Section 2 shows the main variables at cultivation production and post production stage; Section 3 presents the monitoring device, current consumption model, and introduces the primary stages through an overview of our approach to the adaptive sampling period; Section 4 describes results; Section 5 presents the discussion; and Section 6 exposes conclusions and future work.

## 2. Influence of Variables at Cultivation Production and Post-Production Stage

Table 1 shows articles that have worked on the study of environmental variables that affect agricultural products at the pre-production stage of the value chain, such as [27], who expose weather variables, including rainfall, temperature, humidity, and wind are tied with agriculture production, as well as [28], implement a greenhouse production environment measurement and control system, offers an excellent growth condition based on monitoring of temperature and humidity.

Furthermore, at production stage articles [29] identifies changes due to temperature and relative humidity in flavor, texture, shelf life, nutritional attributes, aroma, among others, once the product harvest occurs for different fresh fruits and vegetables. Same as [30] suggests permanent monitoring of crops to minimize risks to face dangerous levels of temperature and humidity.

In the production stage, specifically after harvest, variables, such as concentration of oxygen, light levels, and gases [31,32], are relevant because they generate decomposition of products or materials; however, variables such as temperature and humidity continue to be necessary.

Additionally, several works focus on cold chains in the transport phases, which are transverse to the products in agricultural value chains which represent the main fruits and vegetables susceptible to chill injury variables [31].

In general, throughout the value chain, it is essential to emphasize that the most critical variables are temperature and humidity, so they have to be at limited levels. As evidenced in articles [32,33].

**Table 1 sensors-22-01472-t001:** Documents related to environmental variables in the agricultural value chain.

Ref.	AVC Stage	Environment Variables
[28]	Pre production	Temperature, Relative humidity
[29]	Production	Temperature, Relative humidity
[34]	Production, Post Production	Temperature, humidity
[30]	Pre production, production	Temperature, humidity
[27]	Pre production	Rainfall, temperature, humidity and wind
[35]	Post production	Temperature, humidity
[31]	Transport	Temperature, Humidity, environmental gases
[36]	Distribution	Temperature, Humidity
[37]	Post production, distribution	Temperature
[32]	Post production, distribution	Temperature, humidity, oxygen concentration
[38]	All Stages	Temperature
[33]	All Stages	Humidity

## 3. Energy Consumption Management Approach

This section describes the device selected and current consumption model and introduces the primary stages through an adaptive sampling period method overview.

### 3.1. Monitoring Device

Figure 1 shows the CSCG Tag [39]. It is an IoT solution with an internal processor 32 MHz MCU w/ 512KB internal flash and audio support, which combines features and functions needed for all 2.4 GHz IoT standards into a single SoC (System on Chip). Additionally, it has a holder to install a coin-type battery, a programming port, and radio communication hardware; also, it has ambient light, temperature, humidity, and shock/tilt sensors. The light sensor measures visible light intensity. Simultaneously, humidity and temperature sensors provide high accuracy measurements with very low current consumption in an ultra-compact WLCSP (Wafer Level Chip Scale Package).

### 3.2. Architecture

The IoT devices selected in this study are part of the Smart Farming System based on a three-layered architecture [40]. Figure 2 shows the device belongs to the perception layer located on the farm; they send monitored variables and have communication with a higher layer (Edge Layer) which processes the data and can manage the devices if required. The CSCG label was programmed under C language, the communication configured with IEEE 802.15.4 [41] protocol, conceives a communication range of 10 m with a transfer speed of 250 kbit/s. The devices at the edge layer also receive data from the perception layer, process them, and send it to the servers for further analysis.

### 3.3. Current Consumption

The CSCG tag stays most of the time in low power mode because the minimum sampling periods for environmental variables are in the order of seconds. However, some functions are activated periodically to maintain the state machine and perform sensor reading tasks, data transmission, and commands reception by wireless protocol.

For current consumption analysis, we use the Otii Arc DC Power Analyzer Data Logger device [42] set constant voltage source. Figure 3 shows the device’s current consumption configured for data transmission every minute. The device wakes up every 30 s to update the state machine. Additionally, it shows power-up intervals where peripherals, radio communication, state machine, and configured sensors take about 4 s.

Figure 4 focuses on current consumption when the device is awake; there are five stages, state machine update, sensor readout, transmission, time for reception of configuration commands, and time when the device prepares to go to sleep.

Equation (Equation 1) shows average total current consumed by an IoT device defined as the sum of current consumption multiplied by its duration divided by total time equation.
(1)$$I_{avg}=\frac{I_{awake}*T_{awake}+I_{sleep}*T_{sleep}}{T_{total}}$$
where:$I_{avg}:$ average current consumption (μA);$I_{awake}:$ current consumption when device is awake (μA);$T_{awake}:$ awake time (ms);$I_{sleep}:$ current consumption when the device is on sleep mode (μA);$T_{sleep}:$ sleep time (ms);$T_{total}:$ total time (ms).

Following the power consumption during active periods is:(2)$$I_{awake}*T_{awake}=\lambda *I_{sm}*T_{sm}*ST+I_{sn}*T_{sn}*ST+I_{tx}*T_{tx}+I_{rx}*T_{rx}+I_{sl}*T_{sl}*ST$$
where:ST: times per period of time;λ: number of times the status machine is updated per minute;$I_{sm}:$ current consumption when device is updating the state machine (μA);$T_{sm}:$ state machine time (ms);$I_{sn}:$ current consumption when device is reading sensors (μA);$T_{sn}:$ reading sensors time (ms);$I_{tx}:$ current consumption when device is transmitting data (μA);$T_{tx}:$ transmission time (ms);$I_{rx}:$ current consumption when device is on reception mode (μA);$T_{tx}:$ reception time (ms);$I_{sl}:$ current consumption when device is going to sleep mode (μA);$T_{sl}:$ to go sleep mode time (ms).

The Equation (Equation 2) shows ST is a variable without units that describes the number of times that each stage is executed over the total time; this means that in a period of 8 min, the reading of sensors is performed eight times, during this time the data are stored. When the time is up, the data from sensors are averaged and transmitted, and reception of acknowledge and configuration commands from the central node is expected. Therefore, transmission and reception only take place once in a period.

Following we define: $T_{sleep}$
(3)$$T_{sleep}=\left(T_{total}-T_{awake}\right)$$
where $T_{total}$: corresponds to total time in ms defined as:(4)$$T_{total}=60*1000*ST$$

Replacing $T_{awake}$
(5)$$T_{awake}=\lambda *T_{sm}*ST+T_{sn}*ST+T_{tx}+T_{rx}+I_{sl}*T_{sl}$$

Finally, the equation of current consumption is:(6)$$\begin{array}{c} I_{avg}=\frac{\lambda *I_{sm}*T_{sm}*ST+I_{sn}*T_{sn}*ST+I_{tx}*T_{tx}+I_{rx}*T_{rx}+I_{sl}*T_{sl}}{60*1000*ST}\\ +\frac{\left(I_{sleep}*\left(\left(60000-\lambda *T_{sm}\right)*ST-\left(T_{tx}+T_{rx}\right)\right)\right)}{60*1000*ST} \end{array}$$

Equation (Equation 6) allows determining the average current consumption related to sampling interval and the monitoring device’s performance. Figure 5 shows sampling interval between 1 to 30 min for λ=2.

After developing our energy consumption model for the selected device, the following section introduces our energy consumption optimization management through an overview of the method and a brief description of the algorithm used.

### 3.4. System Overview

Figure 6 shows an method overview for managing optimize energy consumption in IoT monitoring systems based on adaptive sampling period. Overall, the proposed method includes three stages.

*Stage I*: the environment variables are the input of this stage; this shows the variable behavior in the environment where it is installed; the goal is to acquire the follow-up variables with the minimum sampling period allowed. With this, it is possible to identify if a variable presents an excessive sampling because its change is not so fast or requires a lower sampling period. The output of this stage is the monitoring variables dataset.*Stage II*: the variable analysis receives captured data and is responsible for analyzing the variables’ behavior over time-based mainly on the data variance, determining the most appropriate variable to manage energy consumption. The output of this stage is the monitoring variable selected.*Stage III*: starts from a linear equation as a variance function and two unknown constants (α,β); At this point, an iterative process is proposed that begins by defining a combination of values in a defined range, from which a pair of them is taken to test them on the dataset and obtain the evaluation metrics, the iterative process ended when all combinations of values on dataset were tested. Finally, the pair of values that generate the minimum value of the sum of evaluated metrics is selected. Figure 7 summarizes the algorithm flow.

In this study, the evaluation metrics are current consumption equation found in Section 3.3 and mean square error (MSE) defined as:(7)$$MSE=\frac{1}{n}\sum_{t=1}^{n}e_{t}^{2}$$
where:$$e_{t}^{2}={\left(X_{i}-\hat{X_{i}}\right)}^{2}$$
$$X_{i}\to Vector\;of\;observed\;values\;of\;the\;variable\;being\;predicted$$
$$\hat{X_{i}}\to Vector\;of\;predicted\;values$$

With the constant equation values, a monitoring device is programmed which manage the data transmission periods; therefore, the variables and their performance are evaluated concerning fixed sampling techniques.

## 4. Results

### 4.1. Stage I—Data Acquisition

Supracafe [43] is a company that has modelled each stage of the coffee value chain; Figure 8 briefly shows from crop to export stages using Business Process Model and Notation(BPMN); however, we focus only on two value chain stages highlighted in red. For *Crop*, the inputs at this stage are coffee seeds. This stage is associated with four roles of coffee growers at the Crop beginning: Germinator, Seedling, Planting, and Growth. It can take between eight months and two years. It is a stage that takes the longest of the entire value chain. Then, the harvest stage begins, accompanied by transport phases. Coffee beans go through transformation processes, from cherry coffee to parchment coffee and wet to dry coffee. Coffee is classified and threshed in the storage stage, turning into green coffee; this process can take between two weeks and six months. The next stages are land or ship transportation, sale, and export until roasting, where the coffee already acquires all its properties and is ready for consumption. The detailed BPMN-based coffee value chain is available on the following repository [44].

The study case was in the crop and storage stages of the coffee value chain; with IoT monitoring devices installed in the coffee farm “Los Naranjos” we validated the model. This farm belongs to Supracafé, located in La Venta district, in the municipality of Cajibio, Cauca (21–35′08″ N, 76–32′53″ W).

For the crop stage, the devices configured to the lowest sampling period (1 min), three devices in each plot, located at different heights on the tree see Figure 9 and three lots were monitored. The monitoring process was carried out during March, April, June, July, August, and September 2020.

Figure 10 presents the variations in temperature and humidity (RH) in the selected days. The analysis considered days that showed different rainy, sunny, and cloudy climatic conditions. The graph presents changes in temperature with minimum values of 9.42 °C and a maximum value of 35.62 °C. In turn, there are variations in humidity with minimum values of 21.7% RH and a maximum value of 100% RH.

### 4.2. Stage II—Variable Analysis

In this section, we used machine learning tools available in Python; from the collected data analysis by sensors, the main parameters are variable’s variance and error for different fixed sampling intervals.

#### 4.2.1. Temperature Analysis

Figure 11 presents the data captured (red line) and values simulation when changing the sampling period to different fixed values; these values are obtained from an average data real captured. The error presents data sampled every minute. Similarly, an attenuation is observed in the variable’s peak values concerning the original sampling period.

Figure 12 presents variance calculation for fixed sampling periods for the data of every minute. The highest variations in the values are between 9 and 14 h; likewise, there are some minor peaks between 6 to 9 h and 14 to 17 h of the day, although previous results correspond to March 06, this behavior is similar for different days under different climatic conditions.

#### 4.2.2. Humidity Analysis

Figure 13 presents captured data (red line) and a humidity simulation from the values sampled every minute when changing the sampling period to different fixed values and the error it presents concerning the data sampled every minute. It also shows that the selected sampling period’s error is more significant for those given in the temperature graph and saturation in the humidity percentage at night.

Figure 14 presents variance for fixed sampling periods to data sampled every minute. Compared with previous figure, it presents substantial variations between 8 and 20 h with a maximum peak of 29% at 11:30 in the morning. However, variations are much higher than those managed in temperature, making it more relevant for the management system approach.

The previous data corresponds to one day; therefore, we made the same analysis for selected days group.

Figure 15 shows variance distribution to selected days, revealing a recurring variation between 7:00 and 11:30 h, with a peak of variation at 8:30, as well as a second minor peak between 13:30 and 17:00 h. Therefore, based on previous information, implementing an energy management system is proposed, maintaining the characteristics of carrying out sampling at a one minute sampling period and minimizing energy consumption, which manages an inverse relationship concerning variance the error absolute value.

### 4.3. Stage III—Get Equation Values

With selected variable’s variance, this section found the best values for managing the device’s energy consumption from a base equation.

#### Energy Management Equation

Equation (Equation 8) is proposed as a base point for our adaptive algorithm as the variance function and with two constants to find
(8)Fs=α∗Var(t)+β
where: Fs→samplingfrequency
Var(t)→Variancefunction
α,β→unknownsconstants

Subject to the constraints:

$5.555\times 10^{-3}\leq Fs\leq 16.66\times 10^{-3}\to$ sampling frequency range

Equation (Equation 8) is a base equation where the unknown variables are α and β subject to a frequency range equivalent to a sampling period between 1 min and 30 min [45]. The period range is established based on consultation of interested experts to relate changes in humidity and temperature with studies related to production estimation, disease control, storage, transportation, which require a precise estimate of daily average maximum and minimum the variables selected values in this study [46,47,48].

Therefore, based on the equation result, sampling period (Ps) is established by the equation:(9)$$Ps=round\left(\frac{1}{60*Fs}\right)$$

Equation (Equation 9) returns a value based on converting from frequency to time and rounding it to the nearest whole. This value is applied to the next sampling window (next 30 min), where the error is calculated as energy consumption.

We selected two vectors and a day with high variance to define the α and β values. α ranges from 0 to 0.1 with increments of 1 $\times 10^{-3}$, and for β, a range from 0 to 10 $\times 10^{-3}$ with an increment of 0.1 $\times 10^{-3}$. It generates a two-dimensional matrix for each pair of α and β values; thus, we have the error and energy consumption graph.

Figure 16a shows mean squared error (MSE) for different alpha and beta values. The highest error rates occur to values close to or equal to zero in alpha, related to handling fixed sampling time. Through the β variation, it is possible to reduce error since it forces to have lower sampling periods to point that for values greater than 5 $\times 10^{-3}$, the sampling is 1 min corresponding to minimum sampling period and consequently an error of zero.

Figure 16b shows current consumption for different alpha values and beta where a direct relationship is observed for parameters alpha, beta, and energy consumption; if fixed sampling intervals are selected (α=0), the lower current consumption values are obtained.

For simplification purposes, we selected β=0 based on the previous graph due it represents the lowest current consumption curve. The equation is simplified to find a suitable alpha value because both error and current consumption are acceptable minimums. From the elimination of β, the error normalized curves and current consumption is in Figure 17.

Figure 17 shows error(blue line), current consumption (red line), and sum of error and current consumption (green line) in the α function. Error and energy consumption are normalized because the contribution is proportional to the sum function. Due to high error values, the sum function shows a maximum alpha close to zero; and a stable value due to the current function. Likewise, we determined the value to reach the minimum, which corresponds to the lowest consumption under a low error and low current consumption (black vertical line).

We carried out the previous analysis for all selected days. Figure 18 shows MSE and current consumption; they had similar behavior, especially for MSE; current and MSE on Figure 18 are normalized, which generates the general function as shown in Figure 19, where the best α is selected for each day (black lines). As a result, a points distribution shows an alpha value between 2 $\times 10^{-3}$ and 12 $\times 10^{-3}$. We took the mean value for a general equation in the red line.

Therefore, the equation is:(10)$$Fs=4.6\times 10^{-3}*Var\left(t\right)$$
(11)$$Ps=round\left(\frac{1}{60*(4.6\times 10^{-3}*Var\left(t\right))}\right)$$

Finally, we have the next equation:(12)$$Ps=round\left(\frac{3.623}{Var(t)}\right)$$

Equation (Equation 12) defines transmission averaged variables period; it takes as a starting point the variance calculated in a 30 minute window and determines sending period. Ps can be between transmission every minute or a single one in 30 min. Since variance calculation does not involve complex operations, as well as the equation found, can be programmed in the IoT devices firmware, equation does not represent a significant additional time.

### 4.4. Evaluation

To evaluate the equation, we installed two devices at each height of the tree and the evaluation was carried out in one plot, one with lowest sampling period (1 min) and other with the equation’s execution as a function of humidity variance. Figure 20 and Figure 21 show results and demonstrate that our adaptive sampling method is working correctly (These measurements correspond to one day only apart from the whole data-set).

As shown in Figure 22, the IoT device sampling time is 30 min at starting point. The variance equation says humidity is changing, and the measure is updated 12 min after the first measurement window.

Figure 22 shows sampling intervals during measurements. If calculated variance increases, sampling interval becomes smaller. In other words, if there is a significant change in humidity, the IoT monitoring device measures the variables more often. This result proves that our code works fine, and our method can control the sampling period according to variable behavior.

We select a 30 min window because sensors have certain noises, especially humidity measurements, which are not negligible. Since our adaptive sampling method can be susceptible to the input values, noises can result in wrong sampling period adjustment, resulting in an energy loss.

### 4.5. Performance Evaluation

For environmental parameters performance evaluation, we used statistical metrics.

Mean square error (MSE): defined previously in (Equation 7) and parameter base in this study. It can take any positive value with zero indicating a perfect lack of error;Mean absolute relative error (MARE): is defined as ratio of the measurement absolute error to the actual measurement. Relative error indicates how good measurement is relative to the object size being measured. If x is the actual quantity value, x0 is the quantity’s measured value, and δx is absolute error. The relative error is measured (δx)/x;Mean bias error (MBE): this measures the extent to which the estimated value deviates from observed value. It can take any value, with negative values indicating systematic under-estimation and positive values, over-estimation, and zero indicating a perfect lack of bias;Pearson’s correlation coefficient (R) represents a linear dependence between two variables is widely used and easily interpreted, taking a value between −1→1 with one indicating a perfect positive linear correlation [49];Nash–Sutcliffe efficiency coefficient (NSE): is a normalized indicator of model efficiency corresponding to the estimate’s statistical agreement or skill relative to experimental measurements. It takes a value ranging from −∞→1.0, with one being a perfect fit and negative values meaning that the station offers a better estimate.

Table 2 presents results for different metrics evaluated for energy system management. The temperature sensor presents better results than the humidity sensor because the first one has better accuracy (± 0.2 °C) than (± 2% RH) humidity variable register. It was mainly reflected in MSE, which yielded results close to the sensor’s accuracy, although a uniform environment for both devices.

Regarding MARE, the evaluation presents good results because, for this parameter, it considers the size or magnitude of the measured variable; therefore, the humidity values are more significant than temperature, the result yielded a smaller value. Concerning MBE related to a measurement error that remains constant in magnitude for all observations, it shows that temperature and humidity present values close to zero, indicating a pleasing lack of bias. For R, both variables have values close to 1; it indicates that a linear equation perfectly describes the relationship between variables sampled every minute and adaptive sampling. Subsequently, NSE represents that the sampling with adaptive function has a model with good predictive skill.

Above all, current consumption with adaptative sampling was equivalent to a fixed sampling of 8 min, which, based on Figure 5, shows an almost flat part of the curve of device estimated life, this means the percentage decrease is very close to the maximal percentage decrease allowed.

Figure 23 illustrates results obtained under different climatic conditions. There is a more significant variation in temperature and humidity on sunny days; devices adjust to a sampling period of one minute longer than rainy days. The above allows us to evaluate the α parameter behavior and the proposed management system from a lower level of variation in temperature and humidity and more extended sampling periods. We conducted this experiment on conditions presented on different outdoor days where climate variability is more significant than warehouses or transport vehicles.

Finally, Table 3 presents different metrics evaluated for days under different climatic conditions. The results are the approach of the worse scenario, where the system is implemented. The metrics do not present a significant difference concerning selected day analysis that presented great variation. The energy savings compared to the one day analysis decreased by 0.16%.

## 5. Discussion

This section discusses energy consumption management performance compared to results obtained in other proposed models. In summary, adaptive solutions evaluation is mainly compared with fixed sampling schemes, where consumption savings and data quality are primarily considered. In [22], the study selects a suitable sampling frequency during the acquisition process according to signal frequency’s spectral content. This paper obtained energy savings between 44% and 72%, and MSE values between 1.3 $\times 10^{-4}$ and 1.5 $\times 10^{-2}$ for three evaluated signals. Similarly, in [23], the study performs a frequency analysis, mainly through a fast Fourier transform. They found that the algorithm can reduce the number of acquired samples up to 79% concerning fixed sampling frequency. At the same time, generally, mean absolute error (MAE) shows a preservation of the data accuracy. In [20], the study shows a model based on the sensing-driven cluster, correlation-based sampler selection and model derivation and adaptive data collection, and model-based prediction called ASAP. This was compared by introducing two ASAP variants: local approach and central approach under a set of experiments study the performance concerning messaging cost, energy consumption perspective, and the collected data quality. In [26], different aspects were evaluated to highlight their methods conserved energy by 29%, 47%, and 25%, respectively. The last paper has a close proposal since one of the proposed algorithms considers variance; however, the proposed similarity algorithm is different from the strategy proposed in our article.

Our proposal evaluated different metrics by capturing real data, evidencing good results in terms of data quality, such as MSE, MAE, and Pearson correlation. However, it is essential to highlight the effect of variables such as the devices accuracy and sensitivity used. Regarding energy-saving terms, on average, it was a 10.88% significantly low value compared to other related works; however, it corresponds to a value of 89% within the range of possible reduction for the IoT device.

## 6. Conclusions and Future Work

We proposed the adaptive sampling period method to keep the IoT device current consumption to a minimum and maintain the outstanding sensing quality based on MSE of variables, especially for humidity monitoring. We found a specific variance pattern in the everyday humidity and temperature measurements in which humidity is more significant than temperature. It let us reduce transmission, which is a big part of energy consumption on IoT devices. We decided that an adaptive sampling method is appropriate to achieve our goal and develop a technique to adapt the sampling interval of devices based on the variables’ variance, which is a dispersion measure between values in environmental variables.

We simulated the method with Python taking into account a minute-by-minute measured dataset. The outcome proves that an adaptive sampling method decreases transmissions significantly while providing acceptable quality measurements. Finally, we tested our method with real sensors used in the IoT-Agro project, demonstrating the effectiveness and flexibility.

We evaluated the proposed management systems indoors and outdoors; we located two devices, one with sampling every minute and the other with the adaptive system. For temperature evaluation, the mean square error (MSE) was 0.369 °C^2^, a value close to 0.2 °C the accuracy of the sensor; in Pearson’s correlation, the results were above 0.97, in mean absolute relative error (MARE) presented values lower than 0.028.

For humidity, we obtained an error more significant than expected. Despite generating conditions similar to the two devices, the sensor’s accuracy means that two sensors do not deliver the same value under the same conditions. Therefore, the mean square error (MSE) was 1.39% RH, a value lower than 2% RH to the sensor’s accuracy. In terms of Pearson’s correlation, the results gave values above 0.97, in the mean absolute relative error (MARE) presented values lower than 0.03. With the advantages of adaptive sampling, we achieved an equivalent to an 8 minute sampling in terms of current consumption. The lowest admissible sample corresponds to a minute in the hours with a more significant variation.

With the proposed management system, a reduction of 11.04% is achieved; the result is significant because the maximum possible percentage is 12.2%, equivalent to performing sampling every 30 min all the time. The reduction percentage was 90% for the selected day and 89.18% for the selected group of days with different weather conditions concerning the total percentage decrease allowed.

Our adaptive sampling method determines the next sampling period based on the variance of the previous measurements. Thus, if there is considerable noise, it will not affect the result, and since the method has a 30 min window is not very sensitive to a slight change of the measurements; therefore, we can handle the noise. This document shows the best values for α and β by simulations with the humidity variable. However, the target variable can be determined by α and β are user parameters.

From work carried out, opportunities open up to continue improving the proposed energy management system; some current challenges are: To obtain feedback from the system through a forecasting service to fine-tune the selected α parameter; To work in a controlled environment to generate equal conditions for the sensors, obtain good evaluations of developed systems, and perform system evaluations for extended periods to detect faults.

## Figures and Tables

**Figure 1 sensors-22-01472-f001:**
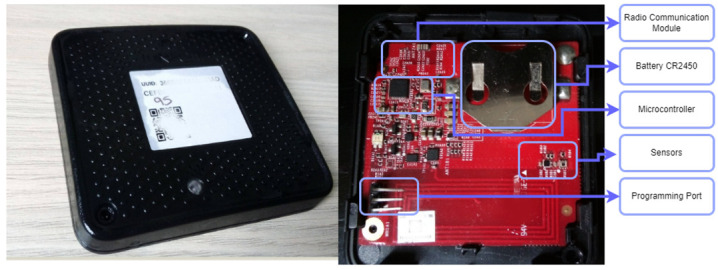
IoT Monitoring CSCG Tag.

**Figure 2 sensors-22-01472-f002:**
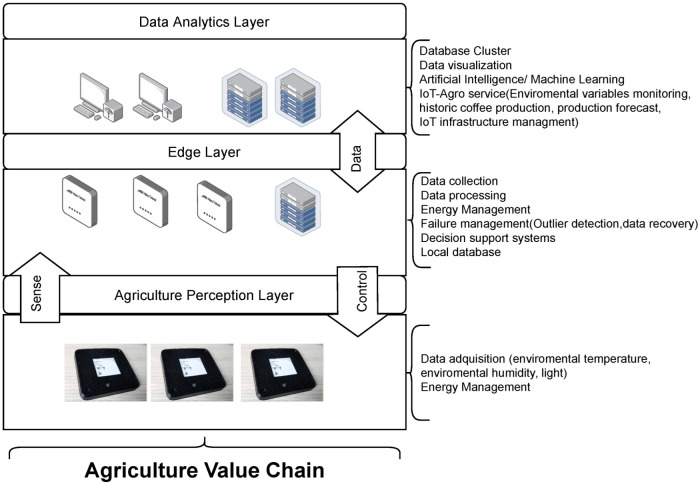
Smart Farming System Architecture.

**Figure 3 sensors-22-01472-f003:**
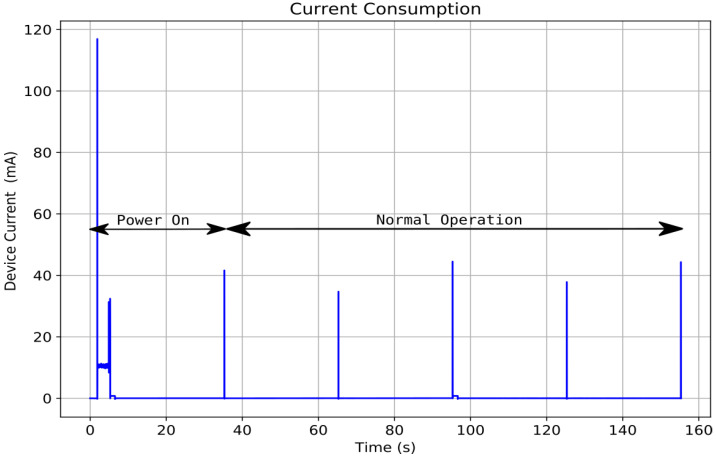
Current consumption on CSCG Tag.

**Figure 4 sensors-22-01472-f004:**
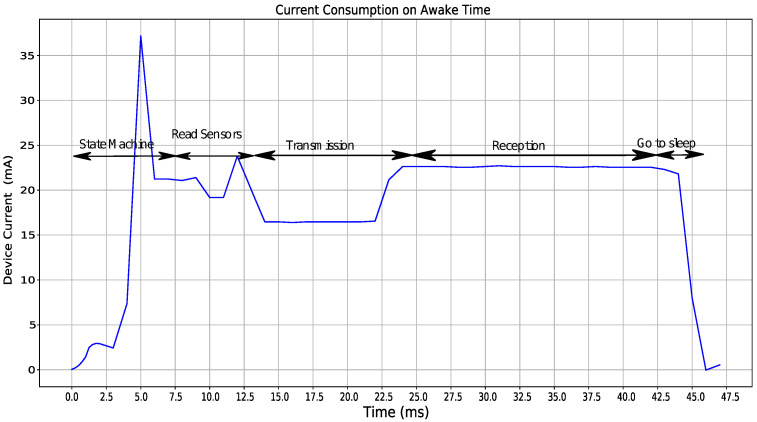
Current consumption during awake time.

**Figure 5 sensors-22-01472-f005:**
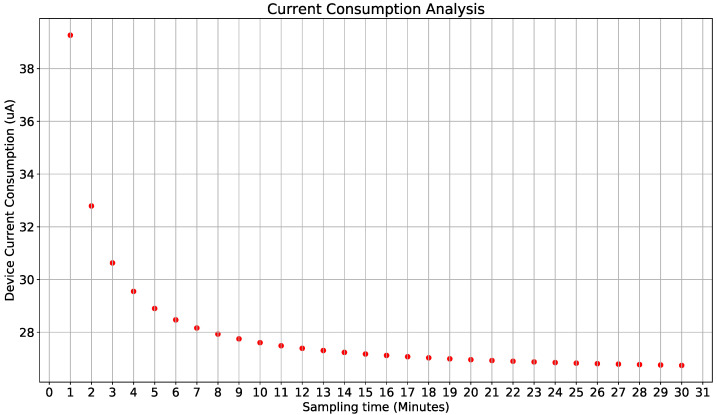
Current consumption vs. sampling interval.

**Figure 6 sensors-22-01472-f006:**
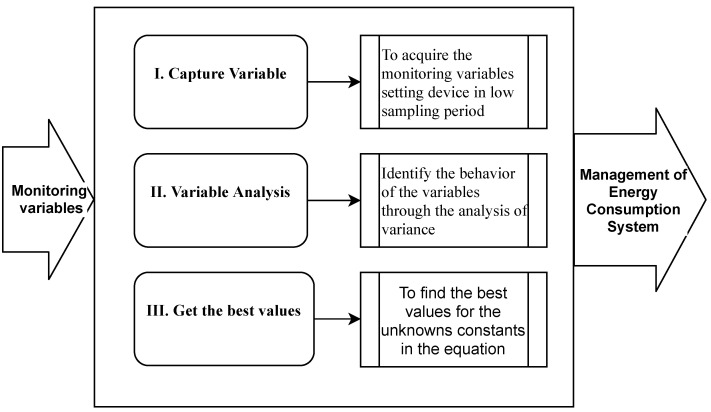
Method for management of energy consumption.

**Figure 7 sensors-22-01472-f007:**
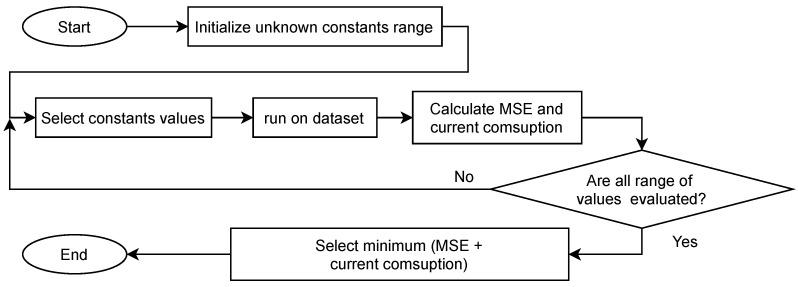
Flow chart to find the best constant values.

**Figure 8 sensors-22-01472-f008:**
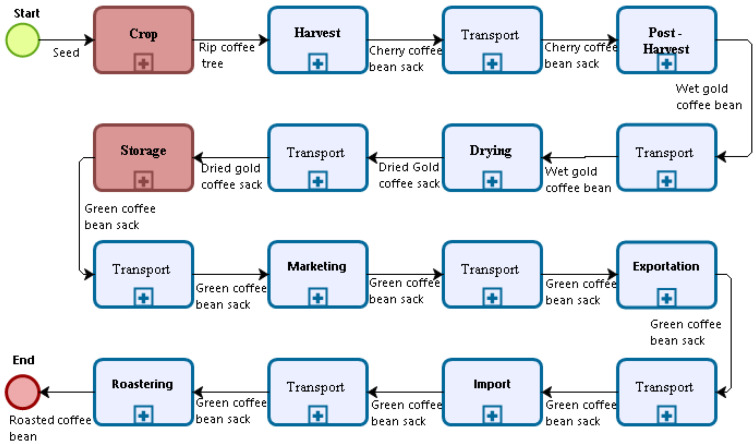
BPMN Coffee value chain.

**Figure 9 sensors-22-01472-f009:**
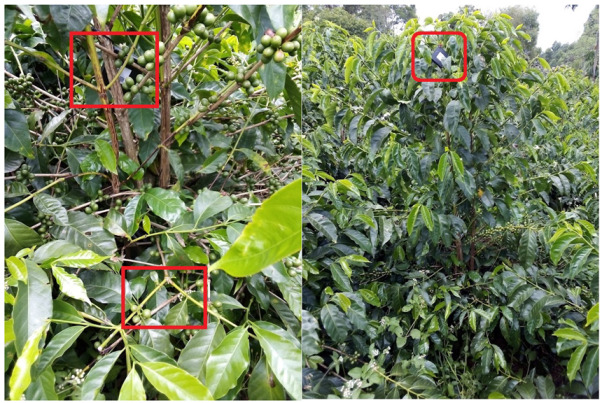
Installation of devices on Coffee plots.

**Figure 10 sensors-22-01472-f010:**
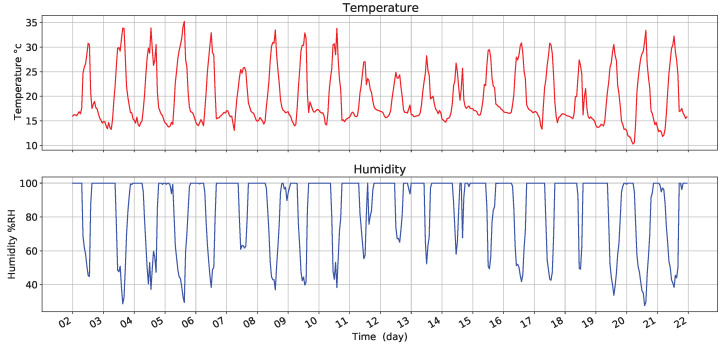
Temperature and humidity at Coffee farm.

**Figure 11 sensors-22-01472-f011:**
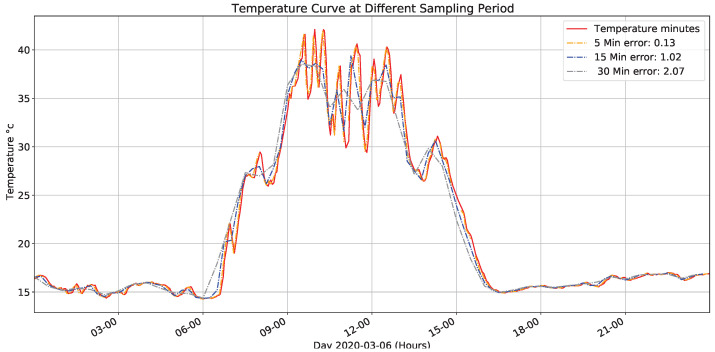
Temperature for different sampling times.

**Figure 12 sensors-22-01472-f012:**
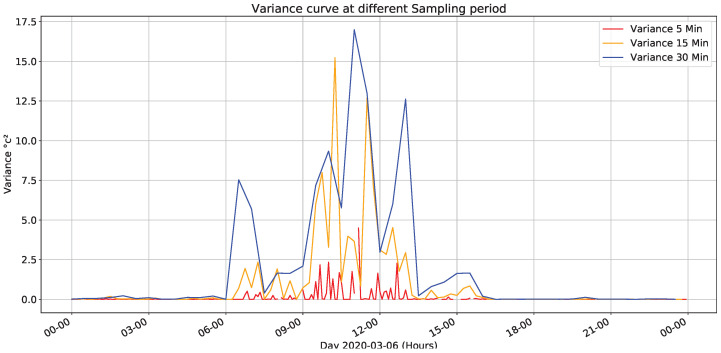
Temperature’s variance for different sampling times.

**Figure 13 sensors-22-01472-f013:**
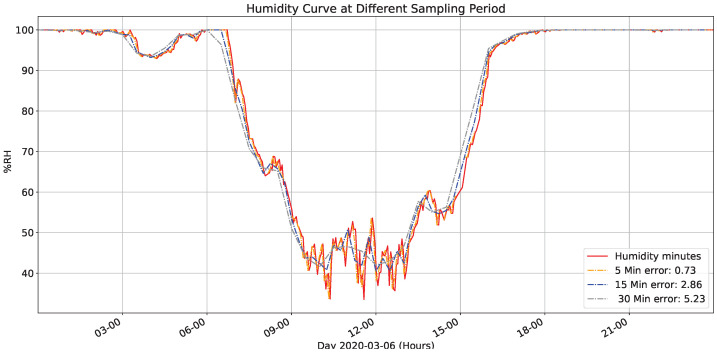
Humidity for different sampling times.

**Figure 14 sensors-22-01472-f014:**
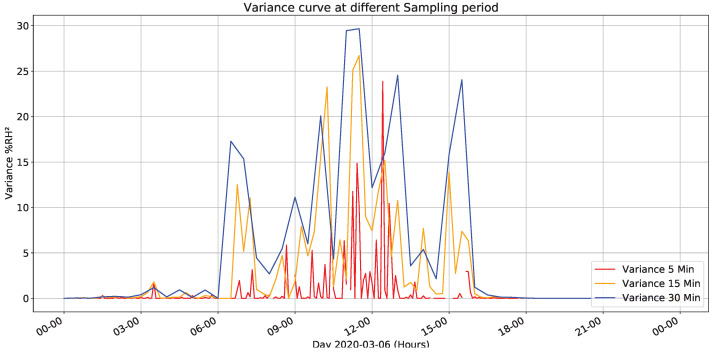
Humidity’s variance for different sampling times.

**Figure 15 sensors-22-01472-f015:**
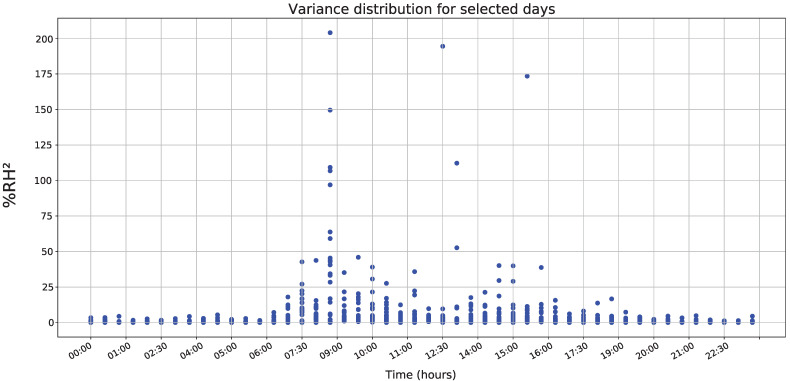
Variance behavior for all datasets.

**Figure 16 sensors-22-01472-f016:**
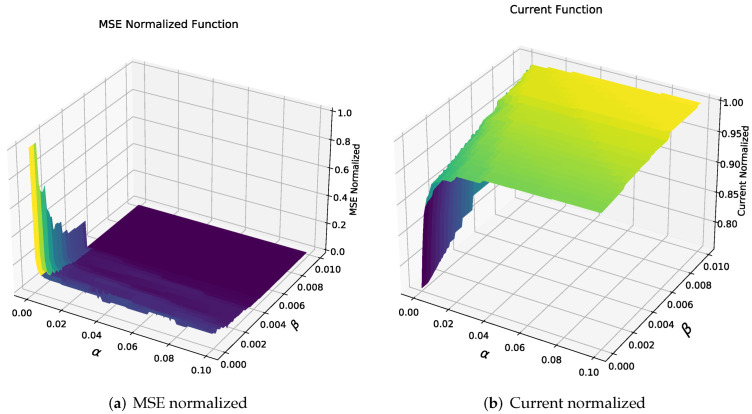
MSE and Current normalized for different α and β.

**Figure 17 sensors-22-01472-f017:**
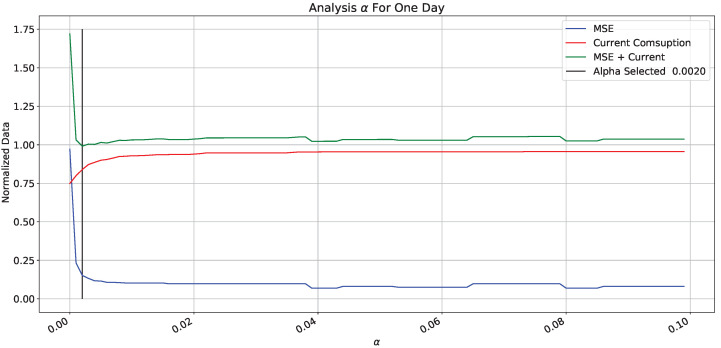
α analysis for one day.

**Figure 18 sensors-22-01472-f018:**
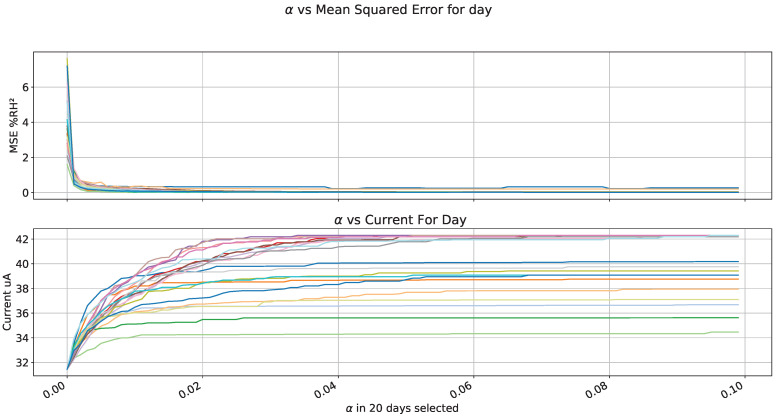
α analysis vs. Mean Squared Error and Current.

**Figure 19 sensors-22-01472-f019:**
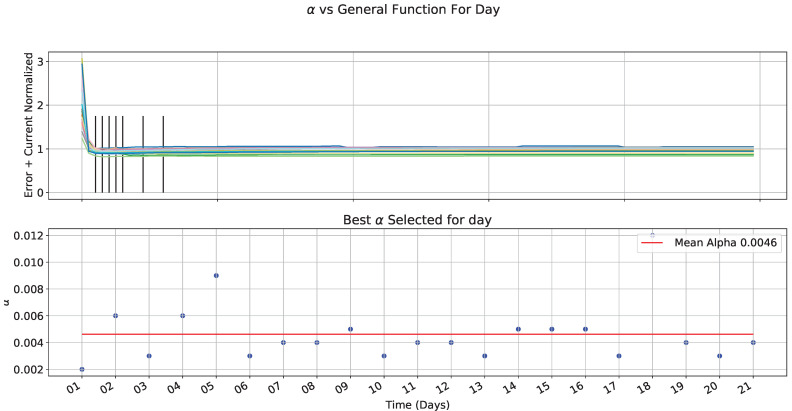
Best α analysis for day and mean α.

**Figure 20 sensors-22-01472-f020:**
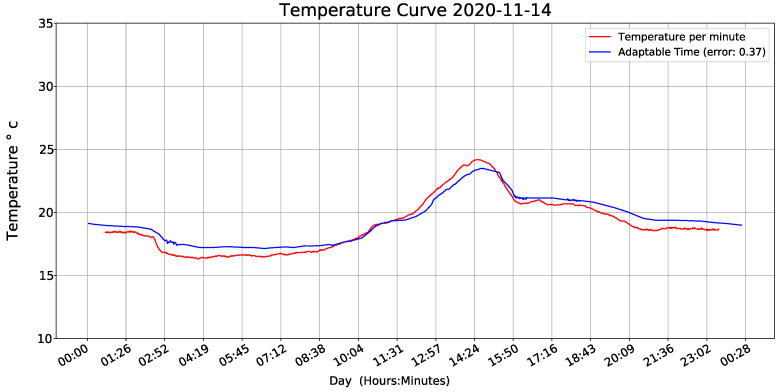
Temperature curve.

**Figure 21 sensors-22-01472-f021:**
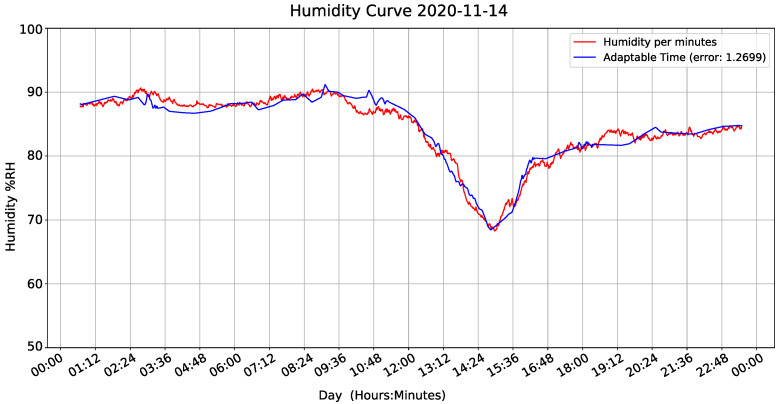
Humidity curve.

**Figure 22 sensors-22-01472-f022:**
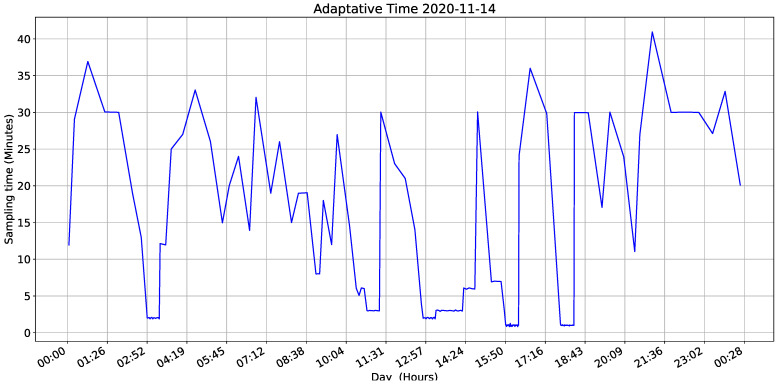
Adaptable time analysis.

**Figure 23 sensors-22-01472-f023:**
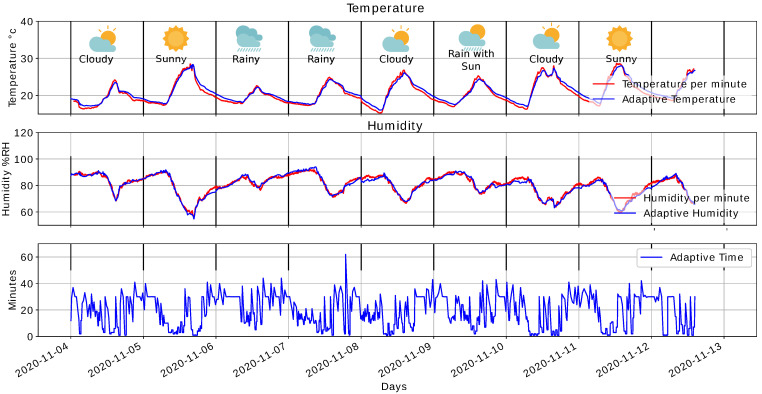
Result for different day conditions.

**Table 2 sensors-22-01472-t002:** Statistics used for evaluation energy system management.

Statistic	Adaptative Time	Range	Ideal Value
MSE Temperature	0.369 °C^2^	0→∞ °C^2^	0.0 °C^2^
MSE Humidity	1.28 $\%RH^{2}$	$0\to \infty \%RH^{2}$	0.0 $\%RH^{2}$
MARE Temperature	0.028	0→∞	0.0
MARE humidity	0.011	0→∞	0.0
MBE Temperature	0.345	−∞→∞	0.0
MBE humidity	0.0015	−∞→∞	0.0
R Temperature	0.979	−1.0→1.0	1.0
R Humidity	0.977	−1.0→1.0	1.0
NSE temperature	0.917	−∞→1.0	1.0
NSE humidity	0.954	−∞→1.0	1.0
Current Consumption	34.92 μA	34.47 μA → 39.27 μA	34.47 μA
% Current Consumption Decrease	11.04%	0% → 12.2%	12.2%

**Table 3 sensors-22-01472-t003:** Resume Statics for days with different conditions.

Statics	Adaptative Time	Range	Ideal Value
MSE Temperature	0.476 °C^2^	0→∞ °C^2^	0.0 °C^2^
MSE Humidity	2.33 $\%RH^{2}$	$0\to \infty \%RH^{2}$	0.0 $\%RH^{2}$
MARE Temperature	0.030	0→∞	0.0
MARE humidity	0.015	0→∞	0.0
MBE Temperature	0.302	−∞→∞	0.0
MBE humidity	−0.25	−∞→∞	0.0
R Temperature	0.983	−1.0→1.0	1.0
R Humidity	0.981	−1.0→1.0	1.0
NSE temperature	0.9498	−∞→1.0	1.0
NSE humidity	0.959	−∞→1.0	1.0
Current Consumption	34.99 μA	34.47 μA → 39.27 μA	34.47 μA
% Current Consumption Decrease	10.88%	0%→12.2%	12.2%

## Data Availability

Not applicable.

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
