# Peer review of "An Adaptive Sampling Period Approach for Management of IoT Energy Consumption: Case Study Approach"

_sensors, 2022, doi:10.3390/s22041472_

Round 1

Reviewer 1 Report

Authors studied the impact on the adaptive Sampling Period Approach on  IoT Energy Consumption using a Case Study in the Colombian Coffee Smart Farm production area so authors can consider changing the title as  An Adaptive Sampling Period Approach for Management of IoT Energy Consumption: Case Study Approach 

Author Response

Response to Reviewer

We want to thank the editor and the anonymous reviewers for their constructive and helpful comments. We have revised the manuscript and applied the suggestions about English and the title

Reviewer 2 Report

The article sounds interesting, addresses practical issues, and includes an extensive list of references, so it may hold the reader's attention. However, there are some minor and serious problems, as described below.

1. Some references lack the website address and/or other publication details, e.g. in [43], [39], [38], [37] and 35].

2. There are minor style errors, such as "Figure 2. Architecture" (whose architecture?), "is expected. therefore" (line 141), "Are All range of..." (Figure 7 - it is not clear whether capital letter is used to emphasize the importance of checking all values), "in the tree. The Figure 9. We carried out" (line 193), "other" is used twice in the same phrase (line 343), "it's important highlight the effects of" (line 360), "possible" is used twice in the same sentence (line 363) making it impossible to understand.

3. What I understood from this article is that Coffee Crop must be monitored, otherwise such a case study would never be discussed or reported. The problem in this regard is that the need to monitor temperature and humidity for Coffee Crop is neither mentioned nor discussed. The authors should justify why monitoring is necessary, even though line 73 talks about minimizing risk "in the face of hazardous temperature and humidity levels." But what should happen when dangerous levels are reached? Do we have to go to the next BPMN step (from Figure 8)?

4. A major shortcoming of this article is that Figure 1 shows an IoT tag, the rest of the article seems to focus on adjusting the sampling rate to minimize power consumption, and Figure 9 shows "installation devices" (although only one is shown in this image and it looks quite different from the one in Figure 1). The problem with this is that although power consumption is of primary interest to this study, the article does not use two or more IoT tags, so no global insight is offered into the power consumption of the whole system. 

5. The basic idea/principle proposed in this paper is adaptive sampling, but this concept is not new. I have worked with ZigBee network implementation and configuring/setting any sampling rate is not a problem. It's true that the ZigBee implementation does not allow adaptive sampling, but if it does, using adaptive sampling for IoT networks should be justified from a practical standpoint (and not necessarily seen as a power consumption requirement).

6. Also from a ZigBee point of view, the corresponding power consumption is quite low and with the button cells currently available on the market, an XBee RF module would work for several years without battery replacement. The question here is why an IoT network is preferred over a ZigBee network, as long as the ZigBee network is specifically designed to manage low transmit power. What about the transmit power of an IoT tag? If it exceeds -40 dBm, an IoT implementation may not make sense, especially if dozens of IoT tags are to be used.

7. As long as the article gives the impression that it is examining a single IoT node, it is not clear how many nodes are required to implement an IoT network with a single router/coordinator. This is an important piece of information as it gives us vital information about the coverage and even the cost of such an implementation. 

8. Figure 3 shows the power consumption measured during normal operation of an IoT tag. In addition, all figures from Figure 10 onwards include statistical data, especially related to the variations of temperature and humidity in a single day. The question that remains here is what happens to the instantaneous power consumption of an IoT tag when the temperature changes from 9.4 to 35.6 degrees. It is likely that power consumption will change with temperature and even the battery will have poorer performance at lower temperatures. 

Based on these comments, I feel that the proposed article lacks confidence and contains very little useful and reliable information.

Author Response

Response to Reviewer

We want to thank the editor and the anonymous reviewers for their constructive and helpful comments. We have revised the manuscript by taking each comment into account. In this document, we explain how we addressed each of the comments and the changes made to the article. 

  1. Some references lack the website address and/or other publication details, e.g. in [43], [39], [38], [37] and 35].

It was a compilation issue, they are solved

  1. There are minor style errors, such as "Figure 2. Architecture" (whose architecture?), "is expected. therefore" (line 141), "Are All range of..." (Figure 7 - it is not clear whether capital letter is used to emphasize the importance of checking all values), "in the tree. The Figure 9. We carried out" (line 193), "other" is used twice in the same phrase (line 343), "it's important highlight the effects of" (line 360), "possible" is used twice in the same sentence (line 363) making it impossible to understand.

They were corrected

  1. What I understood from this article is that Coffee Crop must be monitored, otherwise such a case study would never be discussed or reported. The problem in this regard is that the need to monitor temperature and humidity for Coffee Crop is neither mentioned nor discussed. The authors should justify why monitoring is necessary, even though line 73 talks about minimizing risk "in the face of hazardous temperature and humidity levels." But what should happen when dangerous levels are reached? Do we have to go to the next BPMN step (from Figure 8)?

Coffee cultivation is the case study, the study focuses on providing a solution on the agricultural value chain in general, which is justified in section 2. The devices according to the architecture is in the perception layer, and basically, transmit the information which is processed by the upper layers where alarms and reports are generated to the farmers for decision making.

According to the analysis of the value chain, all stages must be monitored, however, some are more critical than others, therefore more attention is paid to the cultivation phase where due to its prolonged duration it is a challenge in energy terms.

  1. A major shortcoming of this article is that Figure 1 shows an IoT tag, the rest of the article seems to focus on adjusting the sampling rate to minimize power consumption, and Figure 9 shows "installation devices" (although only one is shown in this image and it looks quite different from the one in Figure 1). The problem with this is that although power consumption is of primary interest to this study, the article does not use two or more IoT tags, so no global insight is offered into the power consumption of the whole system. 

The figures were updated

The study is part of a larger study, I attach an article, the article focuses on proposing an adaptive sampling methodology that is programmable in the devices. As such, the whole system has a global management mechanism evaluating other aspects in addition to changing the sampling if the global system requires it.

  1. The basic idea/principle proposed in this paper is adaptive sampling, but this concept is not new. I have worked with ZigBee network implementation and configuring/setting any sampling rate is not a problem. It's true that the ZigBee implementation does not allow adaptive sampling, but if it does, using adaptive sampling for IoT networks should be justified from a practical standpoint (and not necessarily seen as a power consumption requirement).

Correct, this proposal is not linked to the protocol used in the case study, although the consumption of the devices was analyzed, the article aims to present a general methodology based on the sending of values of environmental variables from the variance of the data. From the Zigbee point of view, consumption was reduced by 11% because the protocol and the IoT Tags are quite efficient. We consider that the proposed method is open to be applied under other protocols and sensors where greater reductions in energy consumption are achieved while maintaining the quality of the data.

  1. Also from a ZigBee point of view, the corresponding power consumption is quite low and with the button cells currently available on the market, an XBee RF module would work for several years without battery replacement. The question here is why an IoT network is preferred over a ZigBee network, as long as the ZigBee network is specifically designed to manage low transmit power. What about the transmit power of an IoT tag? If it exceeds -40 dBm, an IoT implementation may not make sense, especially if dozens of IoT tags are to be used.

ZigBee networks are included under the concept of IoT networks. We use the same IEEE802.15.4 protocol that ZigBee uses, and what we seek is to make more efficient use of energy in order to make more intelligent use of the measurement periods according to their variance. There are many configurations for ZigBee depending on the consumption, however, the proposal is not linked to any of them

  1. As long as the article gives the impression that it is examining a single IoT node, it is not clear how many nodes are required to implement an IoT network with a single router/coordinator. This is an important piece of information as it gives us vital information about the coverage and even the cost of such an implementation. 

this information was updated

Results

For the crop stage, the monitoring devices were configured to the lowest sampling period (1 minute), three devices in each plot, located at different heights on the tree see Figure \ref{farmCoffee} and 3 lots were monitored. The monitoring process was carried out during March, April, June, July, August, and September 2020.

Evaluation

To evaluate the equation, we installed two devices at each height of the tree and the evaluation was carried out in one plot

  1. Figure 3 shows the power consumption measured during normal operation of an IoT tag. In addition, all figures from Figure 10 onwards include statistical data, especially related to the variations of temperature and humidity in a single day. The question that remains here is what happens to the instantaneous power consumption of an IoT tag when the temperature changes from 9.4 to 35.6 degrees. It is likely that power consumption will change with temperature and even the battery will have poorer performance at lower temperatures. 

It is true, the batteries have a poorer performance at low temperatures, the results of the study show that during low temperatures that occur at night, there is low variance, therefore the sampling period is increased and the number of transmissions is reduced, This minimizes energy consumption and compensates this behaviour.

Reviewer 3 Report

This paper develops an adaptive sampling period method to save the IoT device energy and maintain the ideal sensing quality based on the variance of variables, particularly for the monitoring of temperature and humidity, which are essential for different stages in coffee AVC. The method was validated in a Case Study in a Colombian Coffee Smart Farm. 

The following points should be considered to improve the paper quality.

  1. In Table 1, the authors list several influencial factors in agriculture, such as Rainfall, temperature, humidity and wind. These factors are well-known by us all. Is there any other factors that are special to agriculture?
  2. Do you consider the correlations among different influential factors? Different factors are often not independent with each other.
  3. Fig.8 is a bit vague. Please replace it with a clearer one.
  4. Proofread the whole paper and correct the existing spelling errors.
  5. IoT has been widely applied to many domains besides agriculture. Therefore, in the Introduction part or Related Work section, I suggest the authors to discuss this issue by adding more related literatures such as the following ones: A Long Short-Term Memory-based Model for Greenhouse Climate Prediction; Artificial intelligence for edge service optimization in internet of vehicles: A survey; Sampling-Based Approximate Skyline Query in Sensor Equipped IoT Networks; IoT-Based Data Logger for Weather Monitoring Using Arduino-Based Wireless Sensor Networks with Remote Graphical Application and Alerts; New Enhanced Authentication Protocol for Internet of Things. Big Data Mining and Analytics.

Author Response

Response to Reviewer

We want to thank the editor and the anonymous reviewers for their constructive and helpful comments. We have revised the manuscript by taking the comment into account.

The following points should be considered to improve the paper quality.

In Table 1, the authors list several influential factors in agriculture, such as Rainfall, temperature, humidity and wind. These factors are well-known by us all. Are there any other factors that are special to agriculture?

Regarding variables, oxygen concentration and exposure to light also affect some stages in particular. However, temperature and humidity  are present throughout the value chain

Do you consider the correlations among different influential factors? Different factors are often not independent with each other.

Correct all the variables are correlated, especially between temperature and humidity, however, the humidity was selected because the variance has more significant changes

Fig.8 is a bit vague. Please replace it with a clearer one.

Fig 8 was updated and it has a better explanation

Proofread the whole paper and correct the existing spelling errors.

The English in the paper was checked again

IoT has been widely applied to many domains besides agriculture. Therefore, in the Introduction part or Related Work section, I suggest the authors to discuss this issue by adding more related literatures such as the following ones: A Long Short-Term Memory-based Model for Greenhouse Climate Prediction; Artificial intelligence for edge service optimization in internet of vehicles: A survey; Sampling-Based Approximate Skyline Query in Sensor Equipped IoT Networks; IoT-Based Data Logger for Weather Monitoring Using Arduino-Based Wireless Sensor Networks with Remote Graphical Application and Alerts; New Enhanced Authentication Protocol for Internet of Things. Big Data Mining and Analytics.

The suggested articles were studied and added, They contribute to the possibilities of the IoT, also are oriented to the security challenge and manage energy consumption through the query technique.

Round 2

Reviewer 2 Report

The article looks better now, but still contains some problems with some images, as stated below:

1. The text on both axes is too small in Figures 4, 15, 21-23.

2. Figure 7 is not vectorized, it looks like it was edited in Paint.

3. The contents of Figures 16 and 17 are too small to be inserted independently, I think they can be merged into a single figure. Otherwise, the impression is that the authors were trying to fill more pages.

Author Response

  1. The text on both axes is too small in Figures 4, 15, 21-23.

The figures were updated

2. Figure 7 is not vectorized, it looks like it was edited in Paint.

The Figure was updated

3. The contents of Figures 16 and 17 are too small to be inserted independently, I think they can be merged into a single figure. Otherwise, the impression is that the authors were trying to fill more pages.

The figures were joined

This manuscript is a resubmission of an earlier submission. The following is a list of the peer review reports and author responses from that submission.

Round 1

Reviewer 1 Report

Please revise/correct the highlighted parts in the paper.

Page 8; Line: 170-171: 'in humidity' should come after variations

Page 9; Line: 183: the date on the text (March 16) does not match with the date in figures (March 06)

Page 12; Line: 221: a and b should be alpha and beta

Page 15; Line: 200-201: 'humidity measurements' is repetitive

Page 18; Line: 326: 'however' is repetitive

Reviewer 2 Report

The authors propose an adaptive sampling period method aimed at being used in the context of IoT systems for environmental monitoring to increase energy efficiency while not affecting the ideal temperature and humidity sensing quality.

The paper is written in a clear and concise style being easy to follow. The organization of the topics and the logical chaining of the sections is in general adequate. The language quality is also appropriate and the paper fits well in the scope of the MDPI Electronics journal.

Regarding the technical content, this reviewer has several concerns. The first one is related with the motivation for the work. Before presenting the idea, the authors need to make a clear and convincing case for addressing the variance of variables such as temperature and humidity, which are very slow changing and can even be easily estimated over time. There is a large body of knowledge regarding this, which should have been considered.

Second, there is no explicit indication  of what are the contributions to the state of the art. This work is a very incremental contribution with limited reach for generalization. 

Third, and highly importantly, there is no actual state of the art. The author's just focus on their implementation without presenting other works and comparatively discuss them comparatively. There is a wide body of knowledge to consider in this work (e.g., "Green Sensing and Communication: A Step Towards Sustainable IoT Systems").

Reviewer 3 Report

No details of the hardware, sensors or communication system used (range, technology) are given.
Line 88 states that it wakes up every 30s, figure 2 shows a significant consumption every 10s.
In figure 3, the consumptions seem very high and it is rare that in transmission it consumes so little (and especially less than in reception), what is the range of the communication technology?

The description and characteristics of the hardware should be described in more detail. Nothing is said about the software. It is also not indicated how and with what instrument the measurements in figures 2 and 3 have been taken.

The expression (2) does not seem to be dimensionally correct, there are terms with time squared (for example, Tsm*ST) and the result is not like that, unless ST is "times per minute" and not "minutes", as indicated in line 104. Something similar happens in the expression (5). Also, I don't understand why the transmission and reception part is not multiplied by ST, isn't the communication done in each cycle? Are the data stored? Communication is the term with the most influence on the consumption with this type of sensors. If indeed expressions (2) and (5) are not correct the rest of the work needs to be changed.

The procedure described in section 3.3 is not clear whether it is performed on a PC or on the device itself.
The procedure, so explained, seems to adjust the error and select the model based on the data taken (say with the learning set), then an evaluation is made (section 4.4 and 4.5) but it is not clear whether the sampling time is fixed or varied automatically (how?) or is varied manually. To better estimate the performance of the procedure you should do a cross-validation and not fix in advance the days for model estimation and the days for evaluation.

Other comments:

- Better to use "Current consumption" than "Power consumption".
- There are too many graphs in section 4.3.
- It seems that the humidity sensors have condensation problems.